# Abnormal oxidative metabolism in a quiet genomic background underlies clear cell papillary renal cell carcinoma

Jianing Xu[1†], Ed Reznik[2,3†*], Ho-Joon Lee[4,5], Gunes Gundem[2,3], Philip Jonsson[1,3], Judy Sarungbam[6], Anna Bialik[6], Francisco Sanchez-Vega[1,3], Chad J Creighton[7,8], Jake Hoekstra[9], Li Zhang[4], Peter Sajjakulnukit[4], Daniel Kremer[4,10], Zachary Tolstyka[4], Jozefina Casuscelli[11], Steve Stirdivant[12], Jie Tang[13], Nikolaus Schultz[1,2,3], Paul Jeng[1], Yiyu Dong[1], Wenjing Su[14], Emily H Cheng[1,6], Paul Russo[15], Jonathan A Coleman[15], Elli Papaemmanuil[2,3], Ying-Bei Chen[6], Victor E Reuter[6], Chris Sander[16,17], Scott R Kennedy[9], James J Hsieh[18], Costas A Lyssiotis[4,19*], Satish K Tickoo[6*], A Ari Hakimi[1*]

[1]Human Oncology and Pathogenesis Program, Memorial Sloan Kettering Cancer Center, New York, United States; [2]Department of Epidemiology and Biostatistics, Memorial Sloan Kettering Cancer Center, New York, United States; [3]Center for Molecular Oncology, Memorial Sloan Kettering Cancer Center, New York, United States; [4]Department of Molecular and Integrative Physiology, University of Michigan, Ann Arbor, United States; [5]Rogel Cancer Center, University of Michigan, Ann Arbor, United States; [6]Department of Pathology, Memorial Sloan Kettering Cancer Center, New York, United States; [7]Human Genome Sequencing Center, Baylor College of Medicine, Houston, United States; [8]Department of Medicine, Baylor College of Medicine, Houston, United States; [9]Department of Pathology, University of Washington, Seattle, United States; [10]Graduate Program in Chemical Biology, University of Michigan, Ann Arbor, United States; [11]Department of Urology, Ludwig-Maximilians University, Munich, Germany; [12]Metabolon Inc, Durham, United States; [13]Genomics Core, Cedars-Sinai Medical Center, Los Angeles, United States; [14]Molecular Pharmacology and Chemistry Program, Memorial Sloan Kettering Cancer Center, New York, United States; [15]Department of Urology, Memorial Sloan Kettering Cancer Center, New York, United States; [16]cBio Center, Dana-Farber Cancer Institute, Boston, United States; [17]Department of Cell Biology, Harvard Medical School, Boston, United States; [18]Department of Medicine, Molecular Oncology, Siteman Cancer Center, Washington University, St. Louis, United States; [19]Department of Internal Medicine, Division of Gastroenterology, Rogel Cancer Center, University of Michigan, Ann Arbor, United States

*For correspondence:
reznike@mskcc.org (ER);
clyssiot@med.umich.edu (CAL);
tickoos@mskcc.org (SKT);
hakimia@mskcc.org (AAH)

†These authors contributed equally to this work

**Abstract** While genomic sequencing routinely identifies oncogenic alterations for the majority of cancers, many tumors harbor no discernable driver lesion. Here, we describe the exceptional molecular phenotype of a genomically quiet kidney tumor, clear cell papillary renal cell carcinoma (CCPAP). In spite of a largely wild-type nuclear genome, CCPAP tumors exhibit severe depletion of mitochondrial DNA (mtDNA) and RNA and high levels of oxidative stress, reflecting a shift away from respiratory metabolism. Moreover, CCPAP tumors exhibit a distinct metabolic phenotype uniquely characterized by accumulation of the sugar alcohol sorbitol. Immunohistochemical staining of primary CCPAP tumor specimens recapitulates both the depletion of mtDNA-encoded proteins

and a lipid-depleted metabolic phenotype, suggesting that the cytoplasmic clarity in CCPAP is primarily related to the presence of glycogen. These results argue for non-genetic profiling as a tool for the study of cancers of unknown driver.

DOI: https://doi.org/10.7554/eLife.38986.001

## Introduction

Cancer cells co-opt normal cellular functions to support malignant proliferation, and in turn offer a glimpse onto the limits of atypical, distorted metabolism. For example, loss of fumarate hydratase (*FH*), succinate dehydrogenase (*SDH*), or isocitrate dehydrogenase (*IDH*) induces pathological accumulation of metabolites which potently inhibit alpha-ketoglutarate-dependent enzymes, inducing hypermethylation of the genome and a fundamental reprogramming of cellular identity (*Wallace, 2012*). Similarly, Otto Warburg observed that tumors take up and ferment large quantities of glucose to lactate in normoxia (*i.e.* aerobic glycolysis), leading him to propose that mitochondrial respiration defects are the underlying basis for cancer (*Warburg, 1956a*; *Warburg, 1956b*). Data from tumor genomics, imaging and metabolomics have now demonstrated that not all tumors exhibit signatures of aerobic glycolysis, and that alterations both intrinsic to mitochondria (*e.g.* mitochondrial DNA (mtDNA) depletion) and extrinsic to mitochondria (*e.g.* increased glucose uptake) can induce aerobic glycolysis (*Vyas et al., 2016*; *Zong et al., 2016*).

While disruption of mitochondrial respiration is generally not viewed as a driver of oncogenesis, a handful of individual cancer types are peculiarly enriched for mitochondrial dysfunction. A recent study described signals of positive selection for loss-of-function somatic mutations in mtDNA in thyroid cancers and chromophobe renal cell carcinomas (chRCC) (*Grandhi et al., 2017*). In parallel, genomic studies have established that many histologies of renal cell carcinoma (RCC) are associated with disruption of mitochondrial respiration and metabolism. For example, the most common RCC histology, clear cell renal cell carcinoma (ccRCC), is characterized by biallelic loss of the von Hippel-Lindau (*VHL*) tumor suppressor and subsequent activation of hypoxia inducible factor (*HIF*), suppressing mitochondrial respiration in favor of glycolytic metabolism. Similarly, chRCCs and renal oncocytomas are characterized by loss-of-function mutations in the mitochondrial genome resulting in accumulation of respiration-defective mitochondria (*Joshi et al., 2015*). Rare subtypes of RCC, such as those driven by biallelic loss of the mitochondrial enzymes *FH* or *SDH*, also potently disrupt normal respiratory function, lending further evidence to the recurrent pattern of mitochondrial dysfunction across RCC histologies.

While a variety of RCC histologies (including clear cell, hereditary leiomyomatosis, and *TCEB1*-mutated subtypes) are characterized by activation of *HIF*, a substantial proportion of *HIF*-activated RCCs harbor no clear genetic cause for *HIF* upregulation. In particular, we and others have described a *HIF*-activated subtype of RCC characterized by cells with clear cytoplasm arranged in a papillary architecture, known as clear cell papillary renal cell carcinoma (CCPAP) (*Rohan et al., 2011*). CCPAP was initially described in kidneys with end stage renal disease (ESRD) (*Tickoo et al., 2006*), but was subsequently also described in patients with normal renal function (*Rohan et al., 2011*; *Gobbo et al., 2008*; *Williamson et al., 2013*). Clinically, CCPAP tumors are of low grade and stage, and are thought to be generally more indolent in nature (*Gobbo et al., 2008*; *Ross et al., 2012*). Morphologically, CCPAP tumor cells are unique in that their nuclei are oriented away from the basement membrane in a linear fashion, and they lack high-grade features including prominent nucleoli, mitotic figures, necrosis and sarcomatoid differentiation (*Gobbo et al., 2008*; *Williamson et al., 2013*). A handful of molecular studies have reported that cytogenetically, CCPAP tumors show neither deletion of chromosome 3p nor mutation of *VHL*, the characteristic molecular features of ccRCC. They also do not exhibit gains of chromosomes 7 and 17, or loss of chromosome Y, as commonly described in papillary renal cell carcinoma (pRCC) (*Rohan et al., 2011*; *Gobbo et al., 2008*; *Williamson et al., 2013*). While several studies have now molecularly profiled CCPAP tumors, no genetic drivers of the disease have been reported (*Rohan et al., 2011*; *Ross et al., 2012*; *Adam et al., 2011*; *Lawrie et al., 2014*).

The unknown genetic basis of CCPAP tumors is a typical example of the large class of molecularly profiled tumors for which no clear driver alteration has been identified. Indeed, a number of studies now describe efforts to characterize the 'long tail' of potential genetic drivers (*Chang et al., 2016*).

Within our own institution's prospective sequencing cohort (*Zehir et al., 2017*), nearly 8% of tumors show no alterations in the 341 cancer-associated genes assayed by the clinical sequencing panel. Here, we report that the outstanding molecular phenotype distinguishing CCPAP tumors from normal tissue (and from other kidney cancers) is predominantly metabolic. Through comprehensive molecular analysis, we find that while CCPAP tumors appear to harbor no recurrent somatic alterations in nuclear DNA, they are broadly characterized by depletion of mtDNA, a drop in mitochondrial RNA expression, and excess levels of sorbitol, glutathione, and NADH. These findings are recapitulated by immunohistochemical staining of primary CCPAP and ccRCC specimens, indicating that in spite of a paucity of evident genetic drivers, CCPAP tumors harbor a distinct metabolic phenotype. This study offers a glimpse onto the pathological consequences of extreme mitochondrial respiratory dysfunction, and may serve as an archetype for understanding tumors with few or no discernable driver alterations.

## Results

### CCPAP is identified as metabolic outlier compared to ccRCC

In our recent metabolomic analysis of ccRCC tumors (herein referred to as dataset RC12) (*Hakimi et al., 2016*), two tumors were identified as metabolic outliers. Specifically, multidimensional similarity analysis using t-distributed stochastic neighbor embedding (t-SNE) indicated that tumors 284 and 159 clustered separately from the 138 other tumors in the RC12 dataset (*Figure 1A*). Pathologic re-review of these metabolic outliers identified them as bearing morphology and immunohistochemical profile characteristic of CCPAP tumors. Interestingly, the primary metabolic feature which distinguished these tumors from the ccRCC samples was the approximately thirty-fold accumulation (relative to levels in adjacent normal kidney) of the sugar alcohol sorbitol.

To confirm if the distinct metabolic features of these outliers could be replicated, we performed metabolomic profiling on an additional cohort of 9 pathologically confirmed CCPAP tumors, eight ccRCC tumors, and 10 normal kidney tissues adjacent to ccRCC tumors. As shown in *Figure 1B*, using t-SNE analysis, CCPAP samples again formed a distinct cluster from conventional ccRCC tumors and from adjacent-normal tissue. Importantly, to evaluate the possibility that sorbitol levels were excessively contributing to the t-SNE analysis, we also included two ccRCC samples from RC12 with aberrantly high sorbitol levels comparable to those from the other RC12 CCPAP samples. We observed that high-sorbitol ccRCC tumors clustered with other ccRCC tumors and not with CCPAP samples. These results were recapitulated using principal components analysis (*Figure 1—figure supplement 1*).

### CCPAP is metabolically characterized by alterations to sorbitol metabolism and oxidative stress pathways

We used the RC13 metabolomic data to interrogate in detail the metabolic changes distinguishing CCPAP from ccRCC. To do so, we calculated the change in metabolite abundance (which we call differential abundance) in CCPAP samples compared to ccRCC samples. After adjusting for multiple-hypothesis testing, we identified 200 differentially abundant metabolites with q value < 0.1 and a greater than 2-fold change relative to ccRCC (non-parametric Mann-Whitney test, *Figure 2A*). Of these, 152 were differentially abundant compared to adjacent normal kidney tissue, confirming the CCPAP-specific nature of these metabolic changes (at least 2-fold change and q < 0.1, Mann-Whitney test). Among these, sorbitol was among the metabolites with the largest change in abundance, exhibiting an approximately 8.5-fold increase in CCPAP samples compared to ccRCC samples (64-fold when excluding the atypical, high-sorbitol ccRCC samples) and a 36-fold elevation compared to normal tissue (*Figure 2A and C*). To confirm the elevation of sorbitol, we independently quantified absolute sorbitol abundance (*Figure 2—figure supplement 1B*) in six tumor samples profiled in RC13 and 3 additional CCPAP tumors not profiled in RC13 (which we referred to as dataset RC15). We observed good agreement between the two quantifications of sorbitol (*Figure 2—figure supplement 1C*. Despite only having six overlapping samples measured in RC13 and RC15, similarly good agreement was observed for the majority of the other commonly measured metabolites (12/27 with Pearson correlation p<0.1, 16/27 with Pearson correlation p<0.2, *Figure 2—figure supplement 1C*). Three other sugar alcohols (palatinitol, erythritol, and ribitol) were also elevated approximately

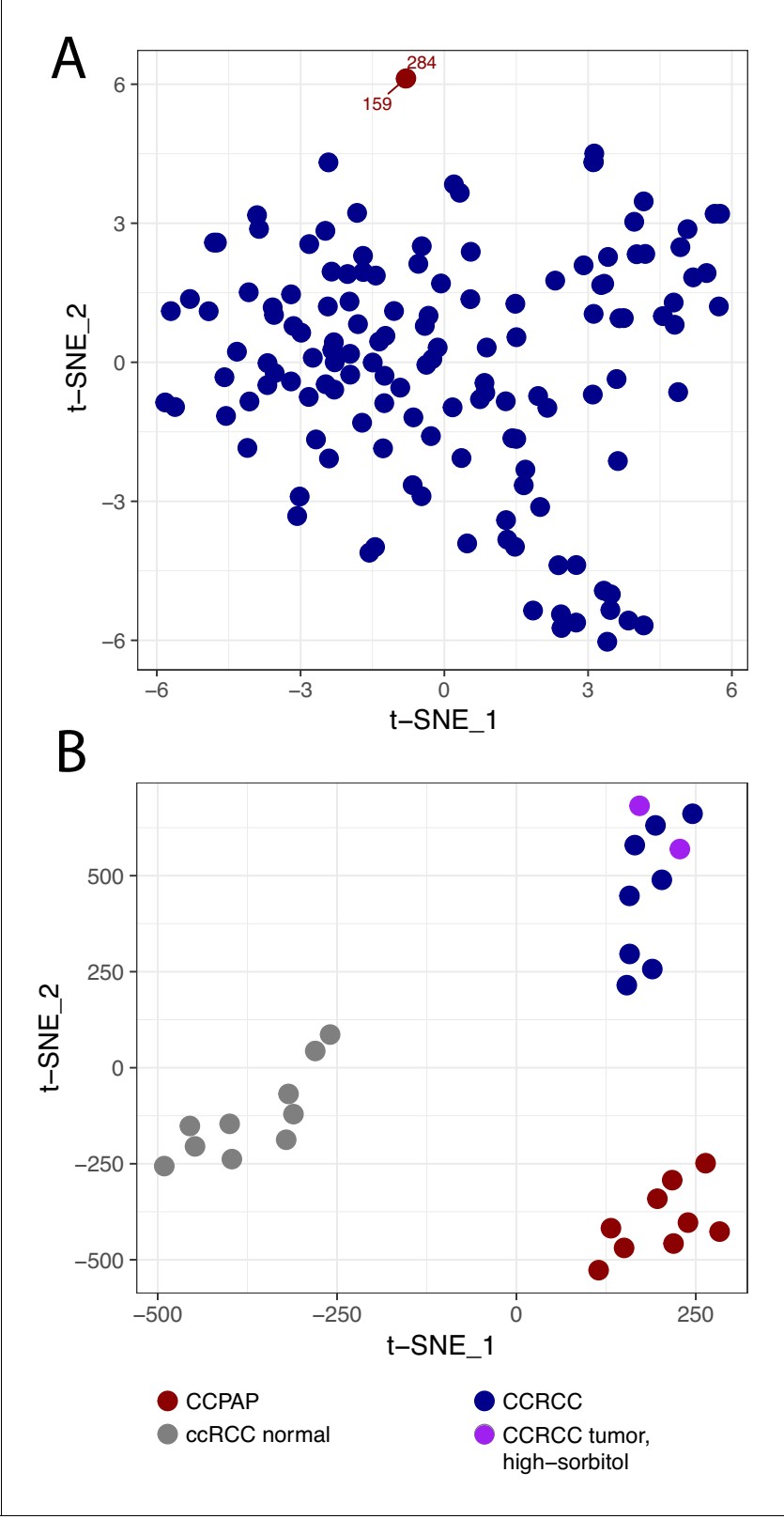

**Figure 1.** CCPAP is a metabolic outlier compared to conventional ccRCC. (**A**) t-distributed stochastic neighbor embedding (t-SNE) metabolomic data from 140 nominal ccRCC identified two outlier tumors (dataset RC12). Pathological review confirmed that these tumors were clear cell papillary renal cell carcinoma (CCPAP). (**B**) t-SNE of metabolomic data of a validation cohort (RC13) of 9 CCPAP and 10 ccRCC tumors confirmed that CCPAP tumors cluster distinctly from ccRCC tumors.

*Figure 1 continued on next page*

*Figure 1 continued*

DOI: https://doi.org/10.7554/eLife.38986.002

The following source data and figure supplement are available for figure 1:

**Source data 1.** Metabolomics results.

DOI: https://doi.org/10.7554/eLife.38986.004

**Figure supplement 1.** Principal components analysis of RC13 data.

DOI: https://doi.org/10.7554/eLife.38986.003

4-fold or greater in CCPAP compared to ccRCC (>5 fold when excluding atypical, high-sorbitol ccRCC samples, *Figure 2—source data 1*), suggesting that a metabolic pathway associated with production or degradation of polyols is differentially active in CCPAP tumors. Importantly, sorbitol could be positively disambiguated from other sugar alcohols of the same mass by virtue of its distinct retention index (*Figure 2—source data 2*). Results of the differential abundance analysis were highly similar when restricting to low stage ccRCC samples in RC13 (*Figure 2—figure supplement 1A*). Under physiological conditions, sorbitol can be produced in response to high levels of osmotic stress. However, other metabolic markers of osmotic stress such as betaine, glycerophosphorylcholine, and myo-inositol, were not increased in CCPAP tumors.

Several other key groups of metabolites were at distinctly higher/lower abundance in CCPAP relative to ccRCC. Notably, CCPAP tumors had significantly elevated levels (>12 fold) of both reduced and oxidized glutathione (GSH and GSSG), the cell's primary anti-oxidants. Increases in abundance of other metabolites involved in the response to oxidative stress were also evident, for example a 4-fold increase in ophthalmate relative to ccRCC (with even greater elevation relative to normal tissue) (*Figure 2A and D*, *Figure 2—source data 1*) (*Soga et al., 2006*). In contrast, a variety of lipids, especially in the class of glycerophosphoethanolamines, were broadly downregulated in CCPAP relative to both ccRCC and normal tissue (*Figure 2A*, *Figure 2—source data 1*).

To further understand the metabolic reprogramming underlying CCPAP, we mapped metabolic changes in CCPAP and in ccRCC (from our earlier study, *Hakimi et al., 2016*) onto a map of central carbon metabolism, augmented with the polyol pathway used to produce/metabolize sorbitol (*Figure 2B*). Doing so exposed a clear distinction between ccRCC and CCPAP metabolism. While ccRCC tumors exhibit elevated levels of several metabolites in upper glycolysis (*e.g.* glucose-6-phosphate (G6P) and fructose-6-phosphate (F6P)), levels of G6P and F6P in CCPAP were not different from normal. In contrast, levels of sorbitol and fructose, the metabolites of polyol pathway, were elevated >30 fold in CCPAP relative to normal tissue. While fructose levels were also elevated in ccRCC (compared to normal), the levels of fructose were 5-fold higher in CCPAP than in ccRCC (*Figure 2B and C*, *Figure 2—source data 1*).

We also observed that the NADH/NAD ratio, a measure of redox potential of the cell and a potential readout of respiratory chain function, was also elevated in CCPAP tumors compared to normal tissue (*Figure 2B and D*, *Figure 2—source data 1*). Determination of the NADH/NAD ratio using metabolomics data is susceptible to artifacts; therefore, we examined our data for other metabolomic changes which may be associated with an elevated NADH/NAD ratio. Using the Recon2 genome-scale human metabolic reconstruction (*Thiele et al., 2013*), we identified all metabolic reactions in the cell dependent on oxidation of NADH/reduction of NAD, and intersected the list of participating metabolites with those measured in our study. We retained all reactions for which we measured all substrates and products. Because several of the central metabolites participating in reactions using NADH/NAD were not quantified in our dataset (e.g. pyruvate, alpha-ketoglutarate, oxaloacetate, dihydroxyacetone phosphate), a number of canonical NAD-dependent reactions were not retained. However, we were left with a list of 5 reactions (related to xanthine metabolism, sorbitol metabolism, ascorbate metabolism, and uracil metabolism). For each of these reactions, we calculated the ratio of the abundances of the more oxidized metabolite to the abundance of the more reduced metabolite (specifically, fructose/sorbitol, uracil/5,6 dihydrouracil, xanthine/hypoxanthine, urate/xanthine, dehydroascorbate/ascorbate). In conditions of elevated NADH/NAD, we would expect the chemical equilibrium to shift these ratios lower. Consistent with this, 4 of 5 metabolite ratios exhibit a drop in CCPAP tumors consistent with an elevated NADH/NAD ratio (i.e. a drop in the ratio). Interestingly, such a drop is not evident in ccRCC tumors with normal sorbitol levels, but is

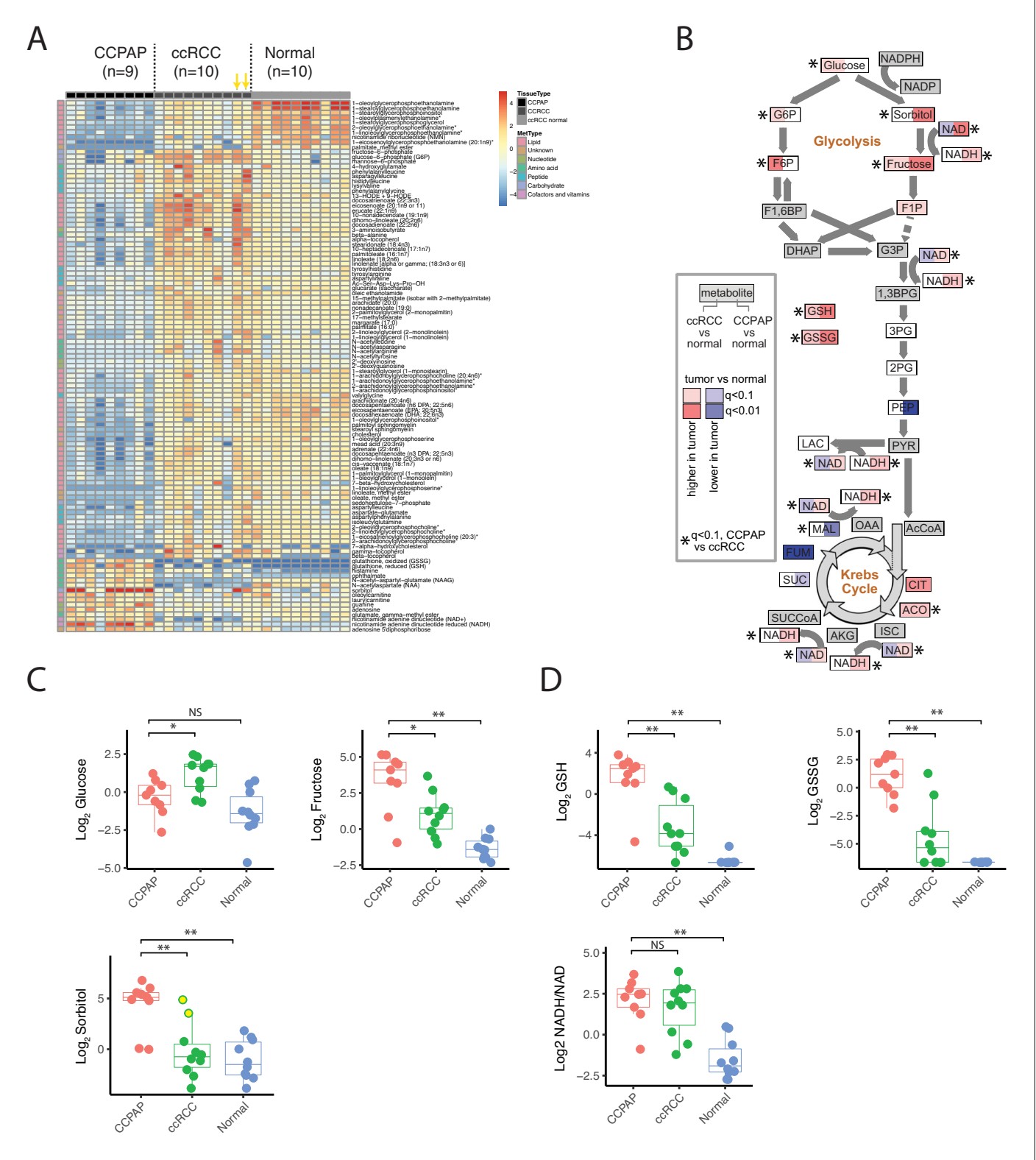

**Figure 2.** Metabolic landscape of CCPAP. (**A**) Heatmap of differentially abundant metabolites among CCPAP (n = 9), ccRCC (n = 10) and matching normal (n = 10). For visual clarity, only metabolites with Mann-Whitney q < 0.1 and absolute log$_2$ fold-change >1 are displayed; a full table of results is available in *Figure 2—source data 1*. Yellow arrows indicate high-sorbitol ccRCC tumors. (**B**) Pathway map depicting metabolite changes (tumors relative to normal kidney tissue) in central carbon metabolism. Metabolites in ccRCC (left half of each box) significantly different from CCPAP (right half of each box) are indicated. (**C**) Relative abundances of key metabolites in CCPAP, ccRCC or normal tissue in polyol pathway. Yellow dots in ccRCC (Log

*Figure 2 continued on next page*

*Figure 2 continued*

Sorbitol plot) indicate high-sorbitol ccRCC tumors. (**D**) Relative abundances of reduced and oxidized glutathione (GSH and GSSG) and NADH/NAD ratio in CCPAP, ccRCC or normal tissue. NS, q > 0.1;*,q < 0.1; **, q < 0.01, Mann-Whitney U test., BH-corrected for multiple hypothesis testing.

DOI: https://doi.org/10.7554/eLife.38986.005

The following source data and figure supplements are available for figure 2:

**Source data 1.** Differential abundance of metabolites.

DOI: https://doi.org/10.7554/eLife.38986.008

**Source data 2.** Metabolite information related to metabolomics data in RC13.

DOI: https://doi.org/10.7554/eLife.38986.009

**Figure supplement 1.** Additional metabolomic analysis of CCPAP tumors.

DOI: https://doi.org/10.7554/eLife.38986.006

**Figure supplement 2.** Comparison of metabolite ratios across different tissue types for metabolites involved in NAD reduction reactions.

DOI: https://doi.org/10.7554/eLife.38986.007

evident in the two sorbitol-high ccRCC tumors (*Figure 2—figure supplement 2*). While metabolite abundance is not a surrogate for metabolic flux, a clear pattern of increases to the constituent metabolites of the polyol pathway (and a concomitant lack of changes to metabolite levels in upper glycolysis) suggests that CCPAP tumors may be preferentially shunt carbon flux to sorbitol, potentially as an approach for regenerating NAD+ and relieving redox imbalance.

## CCPAP exhibits dysfunctional mitochondrial respiration associated with depletion of mitochondrial DNA and RNA

To further understand the molecular mechanisms driving CCPAP pathology, we utilized several cases of misclassified CCPAP which had been submitted to and profiled by The Cancer Genome Atlas (TCGA) consortium in the ccRCC (TCGA ID: KIRC) study. Five suspected cases of CCPAP (BP-4760, BP-4784, BP-4795, DV-5567, BP-4177) were identified during re-review by the TCGA panel pathologists, including two at our institution as part of the TCGA effort and excluded from the final TCGA manuscript (*Figure 3—figure supplement 1*). Importantly, prior to exclusion, each of these tumors was to some extent molecularly profiled, including by RNA-sequencing, whole-exome sequencing, methylation profiling, and miRNA sequencing.

To interrogate the molecular mechanism underlying the metabolic changes evident in CCPAP, we compared gene expression levels between five CCPAP cases, conventional ccRCC tumors, and adjacent-normal kidney tissues in the TCGA (*Figure 4—source data 1*). We identified two sets of differentially expressed genes, which either distinguished CCPAP from adjacent-normal tissue ($g_N$) or distinguished CCPAP from ccRCC ($g_C$). We then determined the pathways enriched for each gene set ($g_N$ and $g_C$) by gene set enrichment analysis (GSEA). GSEA was performed in an unbiased manner on all 4731 gene sets in the C2/curated gene set collection, as well as all 'Hallmark' gene sets, from the MSigDB database (*Subramanian et al., 2005*). Among these, we observed that the most significantly down-regulated gene-sets enriched in either the $g_N$ or $g_C$ lists were associated with oxygen-dependent mitochondrial metabolism (*e.g.*, 'MOOTHA_MITOCHONDRIA' 'HALLMARK_ OXIDATIVE_PHOSPHORYLATION') (*Figure 3C* and *Figure 3—source data 1*). Furthermore, several of the most significantly up-regulated gene-sets enriched in the $g_N$ list of differentially expressed genes were associated with the hypoxic response (*e.g.*, *ELVIDGE_HYPOXIA_UP*, q value < $10^{-5}$) (*Figure 3—source data 1*), consistent with previous findings that HIF is upregulated in CCPAP (*Rohan et al., 2011*).

The analysis above suggested that CCPAP tumors downregulated expression of mitochondrial genes. Mitochondria are heterogeneously composed of proteins encoded in both the 16 kb circular mitochondrial genome (mtDNA) and the conventional nuclear genome (nDNA). While the vast majority of mitochondrially-localizing proteins are encoded in nDNA, there are 13 mtDNA-encoded genes that produce proteins, all of which are essential integral membrane components of the mitochondrial electron transport chain or ATP synthase. Although mtDNA (and mRNAs produced from mtDNA) are highly abundant and typically captured by conventional DNA (often in an off-target manner) and RNA sequencing, analysis of these 13 genes is not commonly reported by the TCGA consortium.

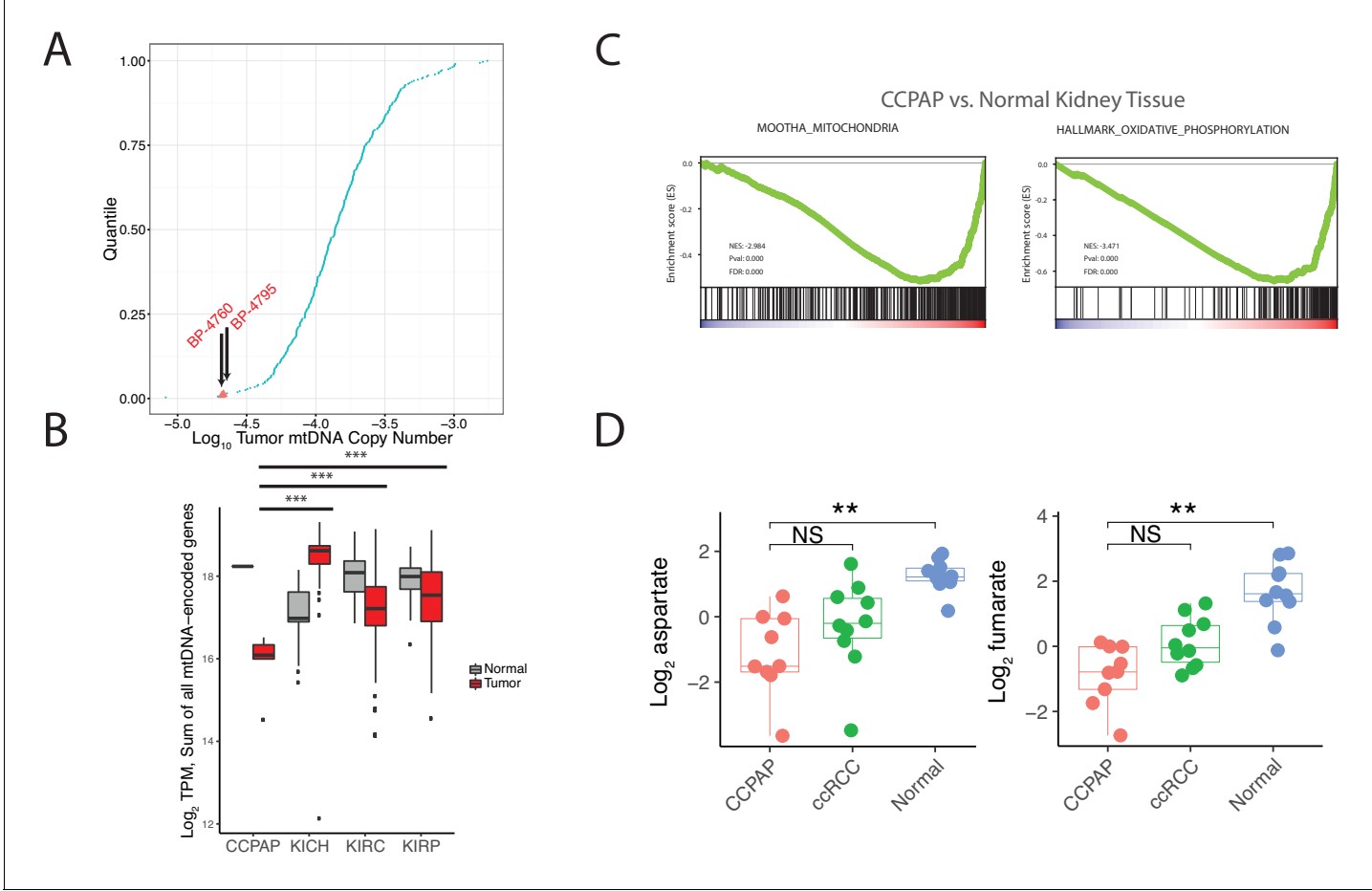

**Figure 3.** CCPAP tumors exhibit suppressed mitochondrial respiration. (**A**) Plots of log transformed mtDNA copy number of 322 TCGA nominal ccRCC samples. Two misclassified samples (BP-4760 and BP-4795) were pathologically re-confirmed as CCPAP. (**B**) Quantification of transcription from the mitochondrial genomes of tumor or normal samples from different subtypes of kidney cancers. ***, p<0.001, *t*-test. (**C**) Gene set enrichment analysis of transcriptome of CCPAP compared with adjacent normal tissue showing that CCPAP tumors exhibit a downregulation of genes related to mitochondria, and oxidative phosphorylation. (**D**) The relative abundance of aspartate and fumarate in CCPAP, ccRCC or normal tissue. NS, q > 0.1; **, q < 0.01, Mann-Whitney U test following multiple hypothesis correction.

DOI: https://doi.org/10.7554/eLife.38986.010

The following source data and figure supplement are available for figure 3:

**Source data 1.** RNA (including miRNA) Sequencing Results.

DOI: https://doi.org/10.7554/eLife.38986.012

**Source data 2.** GSEA results.

DOI: https://doi.org/10.7554/eLife.38986.013

**Figure supplement 1.** H and E image of 5 misclassified TCGA-KIRC samples, which has been re-evaluated to be CCPAP.

DOI: https://doi.org/10.7554/eLife.38986.011

To evaluate the possibility that the mitochondrial genome may be impacted in CCPAP tumors, we re-analyzed mitochondrial DNA sequencing reads available in the TCGA data for CCPAP samples. In prior work, our group estimated mtDNA copy number across ~322 ccRCC samples profiled by the TCGA, including 2 CCPAP samples misclassified as ccRCC. Re-examining this data, we found that the 2 CCPAP tumors were among the four samples lowest in mtDNA copy number (**Figure 3A**, Mann-Whitney p-value 0.016). Depletion of mtDNA copy number in a tumor cell may lead to a drop in mtRNA, and a consequent drop in the capacity of the cell to conduct oxygen-dependent mitochondrial respiration. Therefore, we evaluated whether mtRNA was reduced in 5 CCPAP samples profiled by RNA-sequencing by the TCGA using previously computed mtRNA expression levels from our group (*Reznik et al., 2017*). In agreement with our finding of mtDNA depletion, CCPAP samples

exhibited a > 2 fold reduction in mtRNA levels compared to ccRCC tumor samples (t-test using TPM estimates from *Reznik et al., 2017*, p-value 0.0002, *Figure 3B*) and a > 3 fold reduction in mtRNA levels compared to adjacent-normal tissue in the KIRC study (p-value $10^{-6}$, *Figure 3B*). Analysis using nuclear-DNA encoded, mitochondrially-localizing genes (rather than genes encoded in the mitochondrial genome) yielded similar results: of the 852 genes in the Mitochondrion GO geneset and expressed at a minimal level in CCPAP tumors, 411 were downregulated and 99 were upregulated in CCPAP tumors compared to normal tissues (*Figure 4—figure supplement 1A*). Consistent with the down-regulation of mitochondrial function due to mtDNA depletion, we found when re-visiting CCPAP metabolomics data that the abundance of aspartate, a metabolite whose abundance is dependent on normal function of the mitochondrial respiratory chain for its production, was reduced 3.5-fold in CCPAP relative to adjacent normal tissues (*Figure 3D*). Fumarate, another metabolite typically considered to mitochondria-specific, was also reduced 4-fold in CCPAP relative to normal tissue (*Figure 3D*). Altogether, the evidence of mtDNA and mtRNA depletion suggests that expression of mtDNA-encoded proteins is suppressed in CCPAP tumors.

## CCPAP tumors harbor a largely intact nuclear genome

Given the distinct landscape of metabolic and transcriptomic alterations characterizing CCPAP, we sought to determine if CCPAP tumors were driven by genetic alterations similar to those driving more common kidney cancer types, for example ccRCC or pRCC. To do so, we completed whole-genome sequencing on 5 CCPAP tumors and whole-exome sequencing on 2 CCPAP tumors. Identified somatic variants were pooled with 3 CCPAP tumors profiled by exome sequencing in the TCGA.

The results of the sequencing analysis revealed that CCPAP tumors have a characteristically low mutational burden when compared to ccRCC tumors (*Figure 4A*, *Figure 3—source data 2*), and that all CCPAP tumors were both *VHL* and *TCEB1* wild type. Furthermore, we failed to identify a single gene which was non-synonymously mutated in more than one sample. Notably, several of the non-synonymous mutations in CCPAP tumors were in genes previously associated with oncogenesis (*e.g.* GNAQ), but these mutations were not enriched in genes of any particular pathway or gene set. Finally, using an allele-specific copy number detection algorithm (FACETS) (*Shen and Seshan, 2016*), we found that CCPAP tumors are remarkably 'copy-number quiet,' exhibiting few if any focal copy number alterations and no arm-level copy number alterations (*Figure 4B and C*). Using the whole-genome sequencing data, we also evaluated whether CCPAP tumors exhibited potentially oncogenic structural variants (*e.g.* fusions or rearrangements, see Materials and methods). We identified few genomic rearrangements in the 5 WGS samples (8–12 rearrangements per sample). However, none of these were high-confidence as they were reported by <10 reads and were not associated with a copy number change. Thus, the sequencing data available to us does not support the presence of a recurrent genetic driver in CCPAP tumors.

We were further able to use mtDNA reads from the whole-exome and whole-genome sequencing of confirmed CCPAP samples from our institution to verify the mtDNA depletion phenotype described earlier in the 5 TCGA samples. Using an approach similar to that described in *Reznik et al. (2016)*, we estimated mtDNA copy number in all samples for which we had both tumor and adjacent-kidney normal. Relative to normal kidney, we found that CCPAP tumors are approximately ~10 fold depleted of mtDNA (*Figure 4D*).

It is known that inactivating mutations to genes which are essential for mtDNA replication, maintenance, and biosynthesis (*e.g.* POLG, TK2) can result in severe depletion of mtDNA and cause autosomally recessive heritable disorders (*El-Hattab and Scaglia, 2013*). Using a manually curated list of genes (*Figure 4—source data 1*) known to lead to mtDNA depletion or other mtDNA-associated disorders, we evaluated the possibility that these genes may be somatically inactivated (*i.e.* by truncating or otherwise loss-of-function mutations) in CCPAP tumors. We found no non-synonymous mutations in any gene in our manually curated list. We also examined the possibility that other changes, for example alterations to regulatory regions in the genome, may alter the gene expression of essential mtDNA maintenance genes in CCPAP. Using the results of our differential gene expression analysis, we found no mtDNA maintenance genes which were significantly (q < 0.05) differentially expressed in CCPAP relative to both ccRCC and adjacent-normal tissue.

We further evaluated the possibility that somatic changes to the mitochondrial genome itself, for example mtDNA mutations, may themselves be contributing to the mtDNA depletion phenotype

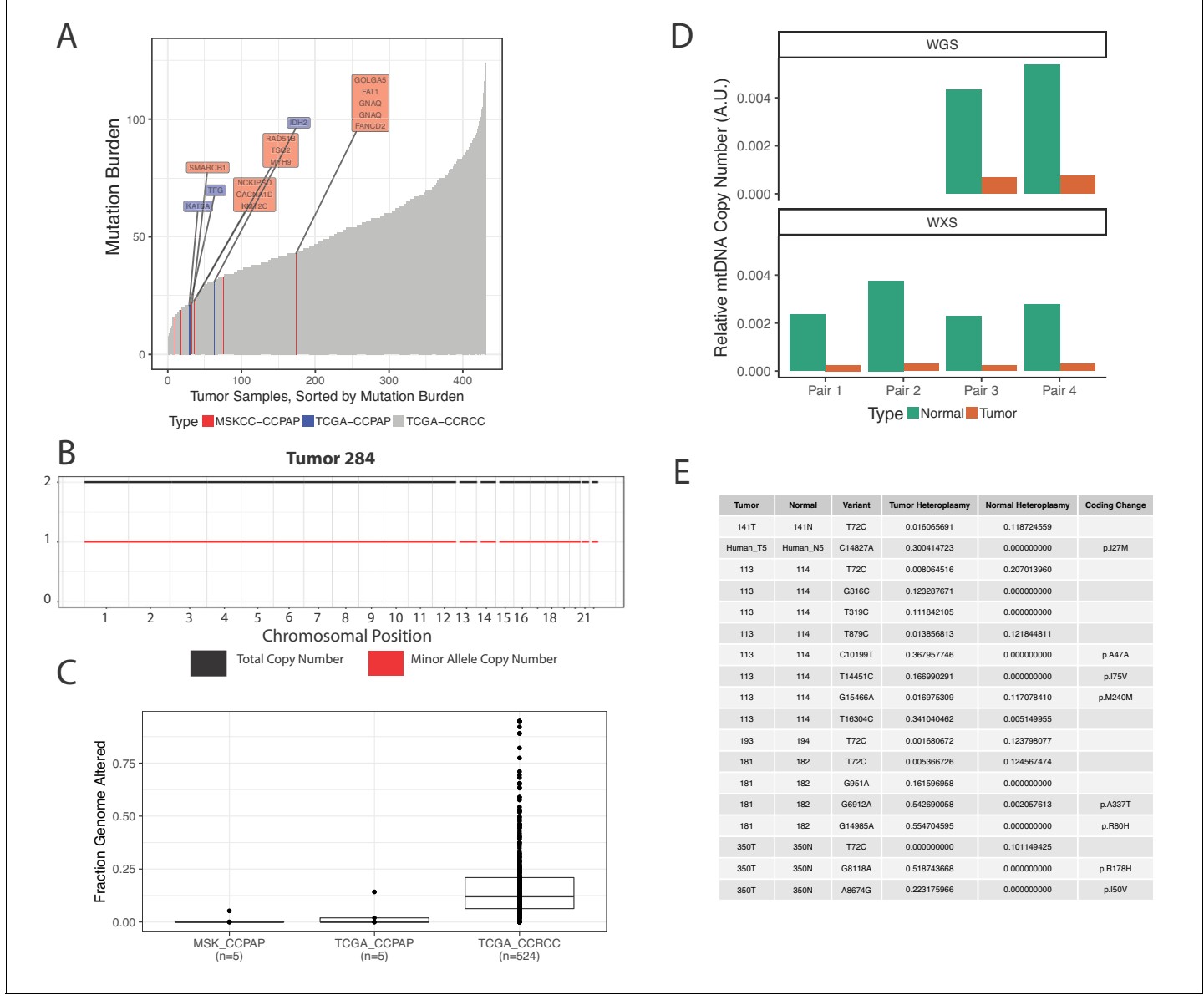

**Figure 4.** CCPAP tumors show few somatic alterations to the nuclear genome, but recurrent depletion of the mitochondrial genome. (**A**) Mutational burden of CCPAP tumors from both MSKCC and TCGA as well as ccRCC from TCGA. Highlighted in the boxes are COSMIC cancer gene census genes which are mutated in either MSKCC (red) or TCGA (blue) CCPAP tumors. Notably, there is not a single gene which is non-synonymously mutated more than once in CCPAP. (**B**) Copy number profile of CCPAP tumor sample 284. (**C**) The fraction of copy-number altered genome from CCPAP tumors profiled by both MSKCC and TCGA, as well as ccRCC tumors profiled by TCGA. (**D**) The relative mtDNA copy number in CCPAP and adjacent-normal tissues that were sequenced by either whole-exome-sequencing or whole-genome-sequencing. (**E**) Somatic variants identified by ultra-deep targeted duplex sequencing of mtDNA of CCPAP tumors.

DOI: https://doi.org/10.7554/eLife.38986.014

The following source data and figure supplement are available for figure 4:

**Source data 1.** Mutation annotation file.
DOI: https://doi.org/10.7554/eLife.38986.016
**Figure supplement 1.** Mitochondrial gene expression analysis of CCPAP tumors.
DOI: https://doi.org/10.7554/eLife.38986.015

we are observing. Using ultra-deep duplex sequencing of mtDNA in CCPAP tumor and adjacent-normal or blood samples (eight total samples), we evaluated whether deleterious/pathogenic mtDNA somatic variants exist in CCPAP samples. As shown in *Figure 4E*, we observed a handful of non-synonymous mtDNA mutations, with a frequency of <1 non-synonymous somatic mutation per sequenced tumor. All identified coding variants were missense mutations, and were not annotated in the literature to have deleterious consequences (using MITOMAP [*Lott et al., 2013*]).

Finally, to evaluate if DNA methylation changes may underlie CCPAP pathogenesis, we examined TCGA data on DNA methylation from 4 CCPAP cases (BP-4784, BP-4795, DV-5567, BP-4177, see Materials and methods, *Figure 5—figure supplement 1B*). Principal components analysis of this data indicated that CCPAP cases did not cluster separately, but instead co-localized with adjacent-normal kidney tissue, rather than ccRCC tumor samples. This suggests that, from a DNA methylation perspective, CCPAP tumors resemble normal kidney tissue.

Taken together, the results above suggest that CCPAP tumors harbor a (likely small) set of cryptic driver alterations which were not identified by our profiling efforts. Such alterations may include, but are not limited to, mutations to enhancer or promoter regions, as well as epigenetic alterations (*e.g.* methylation). Our failure to identify recurrent mutations or copy number alterations is not atypical in the context of our institution's clinical sequencing experience; in fact,~8% of all tumors prospectively deep sequenced for alterations in 341 cancer-associated genes fail to show a single mutation (*Zehir et al., 2017*). Our results so far demonstrate that in spite of the paucity of evident genomic drivers, such tumors can nevertheless manifest with a striking molecular phenotype.

## Immunohistochemical validation of the molecular phenotype of CCPAP

We sought to validate that CCPAP-specific depletion of respiratory genes is not an artifact of TCGA profiling, and to confirm that the depletion of the mitochondrial genome is specific to CCPAP tumor cells, as opposed to other cells in the tumor microenvironment. To do so, we performed immunohistochemical (IHC) staining for mtDNA-encoded MT-CO1, which recognizes the mtDNA-encoded cytochrome c oxidase subunit 1, and for TOM20 which recognizes a nuclear-DNA-encoded mitochondrial import receptor subunit. Consistent with our prior results, we found that the intensity of both TOM20 and MT-CO1 was significantly lower in CCPAP compared with ccRCC and adjacent normal tissue (*Figure 5A*). A more comprehensive quantification using tissue array samples from different subtypes of renal cell carcinoma also confirmed that CCPAP has the lowest MT-CO1 H-score (*Figure 3—source data 2*, *Figure 5B*). In total, this data suggests that CCPAP tumors exhibit highly reduced protein expression of both nDNA and mtDNA-encoded respiratory genes.

Next, we evaluated whether signatures of elevated oxidative stress were evident in CCPAP tumors. High levels of glutathione and opthalmate (a glutathione analog) in CCPAP (relative to normal kidney tissue as well as to ccRCC) suggest the cells have been rewired to combat high levels of oxidative stress. Moreover, microRNA analysis showed that the miR200 family (miR200a,b,c, miR429), which has been shown to be induced by oxidative stress and which suppresses the epithelial-mesenchymal transition (EMT), are upregulated in CCPAP relative to ccRCC (*Figure 5—figure supplement 1A*, *Figure 4—source data 1*), consistent with a previous report (*Lawrie et al., 2014*). To evaluate oxidative stress levels directly, we performed IHC staining of 8-oxo-2'-deoxyguanosine (8-oxo-dG), a marker of oxidative stress, on CCPAP tumor tissue, ccRCC tumor tissue, as well as adjacent normal tissue. We found that the intensity of 8-oxo-dG was higher in CCPAP compared with ccRCC and adjacent normal tissue (*Figure 5C*). A more comprehensive quantification using tissue array samples comprised of different subtypes of renal cell carcinoma also showed CCPAP has the highest 8-oxo-dG H-score (*Figure 5D*, *Figure 5—source data 1*). Importantly, data from our group and others have repeatedly demonstrated that elevation of glutathione is generically observed across many RCC histologies, including but not limited to clear-cell, chromophobe, and oncocytomas (*Hakimi et al., 2016*; *Gopal et al., 2018*; *Priolo et al., 2018*). Our data here suggests that an elevated response to oxidative stress is particularly pronounced in CCPAP tumors, but cannot be used to distinguish them from other histologies.

High levels of 8-oxo-dG may indicate excess DNA damage as well as oxidative stress. Specifically, oxidation of guanine bases is associated with a high prevalence of C:G > A:T transversions, typically associated with Signature 18 in the COSMIC database of mutation signatures. To determine if high levels of 8-oxo-dG in CCPAP tumors may be a readout of oxidative DNA damage, we analyzed the trinucleotide context of somatic mutations in all available CCPAP tumor samples (*Figure 5—figure*

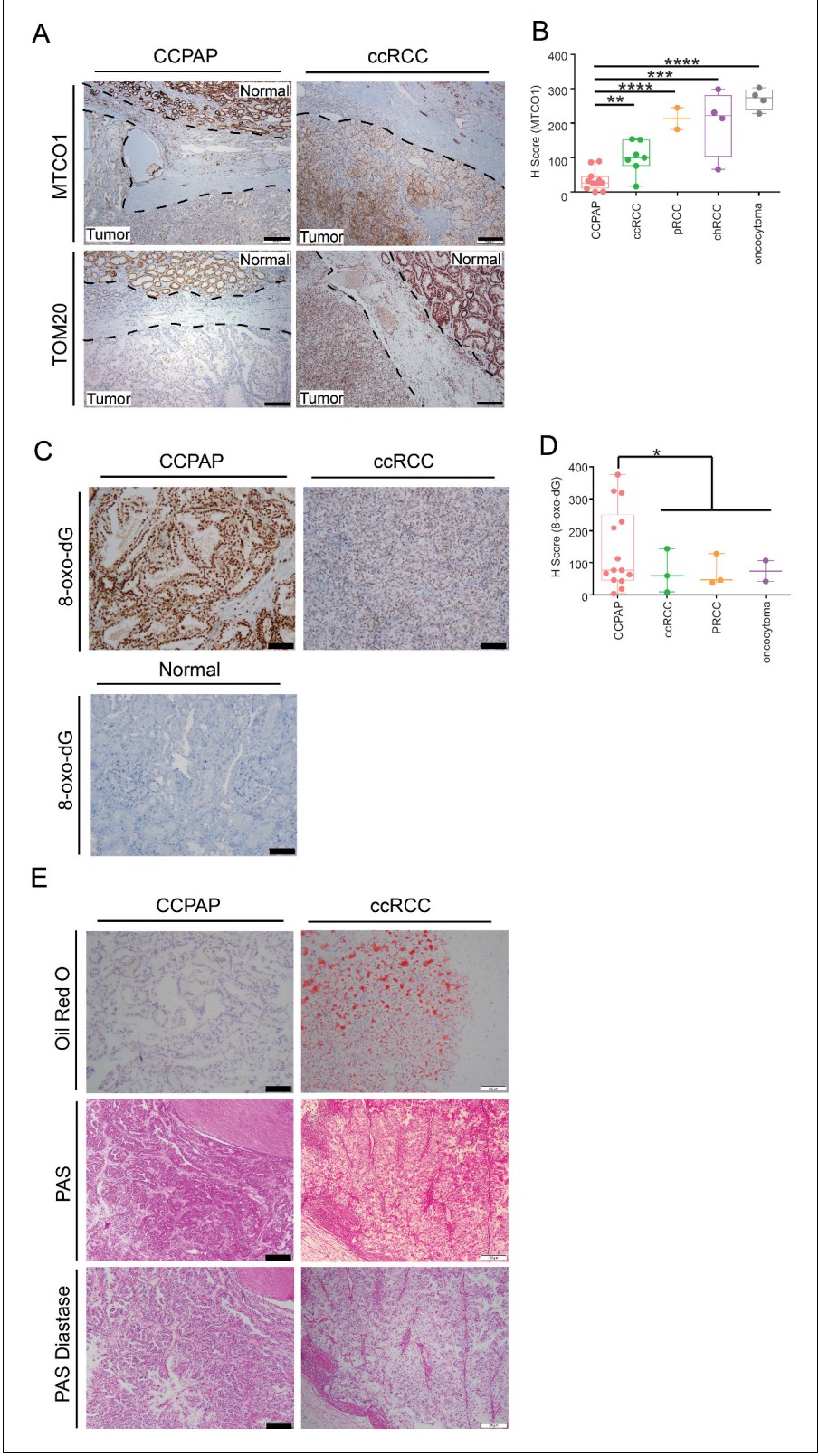

**Figure 5.** Immunohistochemical and histological characterization of CCPAP tumors. (**A**) Representative IHC staining of mtDNA-encoded MT-CO1 and nuclear-DNA-encoded/mitochondrially-localized TOM20 of CCPAP, ccRCC tumors and adjacent normal tissues. (**B**) H-scores of immunohistochemical (IHC) staining for the mtDNA-encoded MT-CO1 protein across different subtypes of kidney cancers (pRCC, papillary RCC; chRCC,

*Figure 5 continued on next page*

*Figure 5 continued*

chromophobe RCC) **, p<0.01, ***, p<0.001, ****, p<0.0001, *t*-test. (CCPAP, n = 11; ccRCC, n = 7; pRCC, n = 2; chRCC, n = 4; oncocytoma, n = 4) (**C**) Representative IHC staining of 8-oxo-dG of CCPAP and ccRCC tumors. (**D**) H-score of IHC staining for 8-oxo-dG across different subtypes of kidney cancers. *, p<0.05, *t*-test. (CCPAP, n = 14; ccRCC, n = 3; pRCC, n = 3; oncocytoma, n = 2) (**E**) Representative images showing the CCPAP and ccRCC tumor regions stained with Oil-Red-O, Periodic acid–Schiff (PAS), and PAS diastase (n = 3 for each staining).
DOI: https://doi.org/10.7554/eLife.38986.017

The following source data and figure supplements are available for figure 5:

**Source data 1.** Score for MT-CO1 IHC and 8-oxo-dG IHC.
DOI: https://doi.org/10.7554/eLife.38986.021
**Figure supplement 1.** MicroRNA and methylation analysis of CCPAP tumors.
DOI: https://doi.org/10.7554/eLife.38986.018
**Figure supplement 2.** Mutation signature analysis of CCPAP tumor samples.
DOI: https://doi.org/10.7554/eLife.38986.019
**Figure supplement 3.** Heatmaps showing intersample correlations (red, positive; blue, negative) between profiles of 5 CCPAP tumors profiled by TCGA as well as average ccRCC (KIRC), chRCC (KICH) and pRCC(KIRP) (rows) and profiles of human and mouse kidney nephron sites (column).
DOI: https://doi.org/10.7554/eLife.38986.020

*supplement 2*). Interestingly, CCPAP tumors did not show an enrichment of C:G > A:T mutations, suggesting that the mutations arising in CCPAP are not the consequence of excess oxidative damage. Together with their low mutation burden, we interpret this to mean that 8-oxo-dG is reporting high levels of oxidative stress which are being sufficiently handled by the cell's stress response (e.g. elevated GSH levels) so as to not generate excess oxidative-stress-associated somatic mutations.

When stained with Haemotoxylin and Eosin (H and E), both CCPAP and ccRCC tumors display a clear cytoplasm. In ccRCC, this clear cytoplasm arsises from accumulated lipid and glycogen contents removed during fixation and processing for histology. To compare the composition of clear cytoplasm in CCPAP to ccRCC tumors, we stained specimens from both tumor types with Oil Red O (for lipids) and PAS and PAS-Diastase (for glycogen). Both CCPAP and ccRCC stained positively for PAS and were sensitive to diastase treatment (PAS-diastase) (*Figure 5E*), indicating the presence of cytoplasmic glycogen in both tumor types. However, unlike ccRCC tumors which stained diffusely positive for Oil Red O, CCPAP tumors are Oil Red O negative (*Figure 5E*). Consistently, the RC13 metabolomics data indicated that CCPAP and ccRCC tumors showed elevated levels of maltose, maltotriose, and maltotetraose (surrogates of glycogen content) compared to normal tissue (*Figure 2—source data 1*). Meanwhile, the RC13 metabolomics data showed that CCPAP tumors are depleted of a variety of lipid classes compared to both ccRCC and adjacent normal kidney tissue (*Figure 2A*). Together, these results indicate that CCPAP tumors do not contain high levels of intra-cytoplasmic lipids, and that the cytoplasmic clarity in CCPAP tumors is due to reasons distinct from those in ccRCC tumors.

## Discussion

In this work, we have described a distinct class of kidney tumors characterized not by their genetics, but by their metabolism. We subjected CCPAP tumors to a battery of comprehensive genomic profiling, much of which was beyond the conventional clinical-grade targeted genetic sequencing offered to cancer patients for identification of driver alterations. With the exception of a single recurrent genotype (mtDNA depletion), no clear pattern of somatic alterations was evident in CCPAP tumors. On the other hand, metabolic analysis readily distinguished CCPAP tumors from their ccRCC counterparts, and identified a striking metabolic disruption in the polyol pathway that defined the metabolic phenotype. While genetic/epigenetic alterations are certainly the drivers of all (or nearly all [*Mack et al., 2014*]) cancers, our data reinforces the notion that such driver alterations need not be the distinguishing molecular feature of the tumor itself. In analogy to the morphology and histology of a tumor, which arise from the tumor's somatic alterations but are fundamentally distinct characteristics for tumor diagnosis and treatment, we propose that metabolism itself may (at least for some tumor types) be equally illuminating (*Peng et al., 2018*; *Reznik et al., 2018*). In particular, it

suggests that the study of cancers of unknown driver (which account for ~8% of all sequenced tumors at our institution) may benefit from holistic (*i.e.* non-genomic) profiling, including but not limited to epigenomic, proteomic, and metabolomic measurements.

The severe respiratory dysfunction in CCPAP argue for a new schema for understanding metabolic reprogramming in renal cell carcinomas in general. As described earlier, each of the common types of RCC are characterized in part by mitochondrial dysfunction: *HIF* activation and suppression of mitochondrial biogenesis in clear cell RCC, *FH* mutations in HLRCC, and mtDNA mutations in chromophobe RCC. Rare subtypes of RCC further support the recurring theme of mitochondrial dysfunction: mitochondrial Complex II dysfunction in SDH-mutant RCC, and loss-of-function mtDNA mutations in renal oncocytomas. Therefore, we propose here a 'mitochondrial spectrum' to describe the metabolism of RCCs, which is defined by the respective quantity and integrity of mitochondria in each RCC subtype (as reflected by the H-score of MT-CO1 shown in *Figure 5B*). At the far left end are CCPAP tumors, severely depleted of mtDNA and respiratory capacity. At the other end are oncocytomas, which accumulate massive mitochondrial mass as a response to pathogenically mutated mtDNA. Resting between these two extremes are the remaining cancer types, beginning with mtDNA-depleted clear cell tumors and proceeding to chromophobe renal cell carcinomas. Looking ahead, it will be interesting to understand how each of these tumors copes with its respective form of mitochondrial dysfunction; that is, if a depletion of mtDNA induces a different metabolic adaptation than a loss-of-function mutation to mtDNA.

While CCPAP tumors exhibit a uniquely high abundance of sorbitol, this should not be misconstrued as an indicator that flux through the polyol pathway has increased. Without the direct measurement of metabolic flux, it is not possible to disambiguate whether increased levels of polyols reflect increased flux through the polyol pathway, or conversely are the consequence of decreased catabolism of metabolites downstream of polyol compounds. However, additional interpretation of the metabolomics data may offer a clue toward resolving this question. Both of the terminal components of the polyol pathway (sorbitol-6-phosphate and fructose) are elevated in CCPAP tumors and correlated to the levels of sorbitol. This suggests that, if the accumulation of sorbitol is due to decreased utilization of metabolites downstream of sorbitol, the 'choke-point' most likely occurs after fructose and/or sorbitol-6-phosphate. Furthermore, studies of metabolomic changes associated with mitochondrial disease indicate that sorbitol accumulation is not a generic consequence of mitochondrial dysfunction, although its elevation has been observed in the blood of patients with progressive external ophthalmoplegia (*Esterhuizen et al., 2017*). Together, this suggests that a combination of factors likely contribute to the accumulation of sorbitol in CCPAP.

It is furthermore notable that although the metabolomic and transcriptomic evidence available is suggestive of a down-regulation of oxidative phosphorylation in CCPAP, these tumors did not exhibit an obvious compensatory up-regulation of glycolysis. Specifically, lactate levels in CCPAP tumors were not elevated relative to normal tissue or ccRCC, nor were any other intermediates of glycolysis (*Figure 2*, *Figure 2—source data 1*). If glycolytic flux is indeed not increased in compensation, this may ultimately reflect the indolent and slow-growing nature of these tumors.

Different histologies of renal cell carcinoma are believed to arise from distinct cells in the human nephron. On the basis of gene expression data, ccRCC and papillary RCC (pRCC) are believed to derive from proximal tubule cells, while chromophobe RCC (chRCC) is believed to arise from the distal tubule. Because CCPAP is rare and slow-growing, no commercial CCPAP cell lines are available as models. To infer the cell-of-origin of CCPAP, we compared the transcriptomic profiles (from the TCGA) of ccRCC, pRCC, chRCC, and CCPAP with transcriptional profiles of microdissected regions of mouse kidney (*Cheval et al., 2012*). Doing so, we found that while ccRCC and pRCC tumors most resemble cells of the proximal tubule and chRCC tumors resemble cells of the distal tubule, CCPAP tumors resemble cells of the collecting duct (*Figure 5—figure supplement 3*). This is consistent with prior immunohistochemical findings showing that CCPAP tumors stain positive for the collecting duct marker 34BE12 (*Cheval et al., 2012*). Based on these findings, future efforts at developing in vitro models of CCPAP may benefit from use of collecting duct cells. It will be of particular interest to determine if the unique activity of the polyol pathway evident in CCPAP tumors is reflected in the metabolism of the collecting duct.

An interesting aspect of CCPAP RCC is that despite wild-type *VHL* and *TCEB1*, CCPAP tumors display upregulated *HIF1* protein as well as elevated levels of downstream HIF targets (*CA9*, *GLUT1*) by IHC staining (*Rohan et al., 2011*). Consistently, our gene set enrichment analysis comparing

CCPAP and normal tissues also show that the hypoxic response is among the most upregulated pathways (*Figure 3—source data 1*). Meanwhile, CCPAP tumors displayed biomarkers indicative of elevated levels of oxidative stress compared with CCRCC or normal tissues. Conceivably, high levels of oxidative stress could inhibit prolyl hydroxylase (PHD) activity to prevent the hydroxylation of HIF and subsequent VHL-dependent degradation (*Mansfield et al., 2005*; *Pan et al., 2007*), which would be responsible for HIF activation in CCPAP. Other, previously undescribed mechanisms may also be the source of HIF activation in CCPAP. Interestingly, both HIF activation and oxidative stress have been shown to cause mtDNA depletion (*Zhang et al., 2007*; *Mansouri et al., 1999*; *Chen et al., 2001*; *Suliman et al., 2003*; *Shokolenko et al., 2009*). Therefore, it is possible the high level of oxidative stress and the consequent HIF activation in CCPAP could cooperatively induce mtDNA depletion.

# Materials and methods

## Key resources table

| Reagent type (species) or resource | Designation | Source or reference | Identifiers | Additional information |
|---|---|---|---|---|
| Antibody | Rabbit Anti-TOM20 Antibody | Santa Cruz Biotechnology | Cat# sc-11415, RRID:AB_2207533 | IHC (1:100) |
| Antibody | Mouse Anti-MTCO1 Antibody | Abcam | Cat# ab14705, RRID:AB_2084810 | IHC (1:2000) |
| Antibody | Mouse Anti-8-Oxo-dG Antibody | Genox Corpooration | Cat# MOG-020P, RRID:AB_1106819 | IHC (1:400) |

## Sample acquisition and preparation for sequencing

Human whole exome sequencing library preparation and sequencing 51 Mb. Libraries were prepared using the Agilent SureSelect XT Target Enrichment System in accordance with the manufacturer's instructions. Briefly, 1500 ng of DNA was sheared using a Covaris LE220 sonicator (adaptive focused acoustics). DNA fragments were end-repaired, adenylated, ligated to Illumina sequencing adapters, and amplified by PCR (using six cycles). Exome capture was subsequently performed using 500–750 ng of the DNA library and the Agilent SureSelectXT V4 Human All Exon capture probe set. Captured exome libraries were then enriched by PCR (using 10 cycles). Final libraries were evaluated using fluorescent-based assays including PicoGreen (Life Technologies) or Qubit Fluorometer (invitrogen) and Fragment Analyzer (Advanced Analytics) or BioAnalyzer (Agilent 2100), and were sequenced on an IlluminaHiSeq2500 sequencer (v4 chemistry, v2 chemistry for Rapid Run) using 2 × 125 bp cycles. Normal and tumor samples were sequenced a depth of 80X and 150X, respectively.

WGS library preparation and sequencing, Truseq PCR-free (450 bp). Whole genome sequencing (WGS) libraries were prepared using the Truseq DNA PCR-free Library Preparation Kit in accordance with the manufacturer's instructions. Briefly, 1 ug of DNA was sheared using a Covaris LE220 sonicator (adaptive focused acoustics). DNA fragments underwent bead-based size selection and were subsequently end-repaired, adenylated, and ligated to Illumina sequencing adapters. Final libraries were evaluated using fluorescent-based assays including qPCR with the Universal KAPA Library Quantification Kit and Fragment Analyzer (Advanced Analytics) or BioAnalyzer (Agilent2100). Libraries were sequenced on an Illumina HiSeq X sequencer (v2.5 chemistry) using 2 × 150 bp cycles. Normal and tumor samples were sequenced a depth of 40X and 80X, respectively.

## Alignment

Short insert paired-end reads were aligned to the GRCh37 reference human genome with 1000 genomes decoy contigs using BWA-mem (*Li and Durbin, 2010*).

## Somatic mutation calling

### Substitution

Single base substitutions were called using CaVEMan (Cancer Variants through Expectation Maximisation) (http://cancerit.github.io/CaVEMan/). As described previously (*Nik-Zainal et al., 2012*), the

algorithm compares sequence data from each tumour sample to its own matched non-cancerous sample and calculates a mutation probability at each genomic locus. Copy number and cellularity information for CaVEMan were predicted with the Battenberg algorithm (*Nik-Zainal et al., 2012*) using 1000 Genomes (*Abecasis et al., 2012*) loci within the NGS data. To improve specificity, a number of post-processing filters were applied as follows:

1. At least a third of the alleles containing the mutant must have base quality >= 25.
2. If mutant allele coverage >= 10X, there must be a mutant allele of at least base quality 20 in the middle 3rd of a read. If mutant allele coverage is <10X, a mutant allele of at least base quality 20 in the first 2/3 of a read is acceptable.
3. The mutation position is marked by <3 reads in any sample in the unmatched normal panel.
4. The mutant allele proportion must be >5 times than that in the matched normal sample (or it is zero in the matched normal).
5. If the mean base quality is <20 then less than 96% of mutations carrying reads are in one direction.
6. Mutations within simple repeats, centromeric repeats, regions of excessive depth (https://genome.ucsc.edu/) and low mapping quality were excluded.

Additional unmatched normal filtering was performed using a set of unmatched normal samples. Mutations that were detected in >5% of the unmatched normal normal panel at >= 5% mutant allele burden were excluded.

Variant annotation was done in Ensembl v74 using VAGrENT (*Menzies et al., 2015*).

## Small insertions and deletions

Small somatic insertions and deletions (indels) were identified using a modified version of Pindel (https://github.com/cancerit/cgpPindel) (*Raine et al., 2015*). To improve specificity, a number of post-processing filters were applied that required the following:

1. For regions with sequencing depth <200X, mutant variant must be present in at least 8% of total reads.
2. For regions with sequencing depth >= 200X, mutant variant must be present in at least 4% of total reads.
3. The region with the variant should have <= 9 small (<4 nucleotides) repeats.
4. The variant is not seen in any reads in the matched normal sample or the unmatched normal panel.
5. The number of Pindel calls in the tumour sample is greater than four and either:
    a. The number of mutant reads mapped by BWA in the tumour sample is greater than 0 or
    b. The number of mutant reads mapped by BWA in the tumour sample is equal to 0 but there are no repeats in the variant region and there are reads mapped by Pindel in the tumour sample on both the positive and negative strand.
6. Pindel 'SUM-MS' score (sum of the mapping scores of the reads used as anchors)>=150

Additional unmatched normal filtering was performed using a set of unmatched normal samples (n = 221). Mutations that were detected in >1% of the unmatched normal normal panel at >= 1% mutant allele burden were excluded.

Variant annotation was done in Ensembl v74 using VAGrENT (*Menzies et al., 2015*).

## Structural rearrangements

Structural rearrangements were detected by an in house algorithm, BRASS (Breakpoints via assembly) [https://github.com/cancerit/BRASS], which first groups discordant read pairs that span the same breakpoint and then using Velvet de novo assembler (*Zerbino and Birney, 2008*) performs local assembly within the vicinity to reconstruct and determine the exact position of the breakpoint to nucleotide precision.

## Copy number changes

Segmental copy number information was derived for each sample using the Battenberg algorithm as previously described (*Nik-Zainal et al., 2012*). Briefly, the algorithm phases heterozygous SNPs with use of the 1000 genomes genotypes as a reference panel. The resulting haplotypes are corrected for occasional errors in phasing in regions with low linkage disequilibrium. After segmentation of the

resulting b-allele frequency (BAF) values, t-tests are performed on the BAFs of each copy number segment to identify whether they correspond to the value resulting from a fully clonal copy number change. If not, the copy number segment is represented as a mixture of 2 different copy number states, with the fraction of cells bearing each copy number state estimated from the average BAF of the heterozygous SNPs in that segment. The Battenberg algorithm could not be applied to chromosome X since BAFs are uninformative for male subjects. For this chromosome, logR values were segmented and segmented logR values were converted to copy number estimates as described previously (*Van Loo et al., 2010*).

Without application of the Battenberg algorithm, the resolution of subclonal copy number states is not possible, so copy number segments are called as single integer values (corresponding to the copy number state of the dominant cancer clone) on chromosome X.

## Mutational signature analysis

Mutational signature analysis of the substitutions was performed using the R package DeconstructSigs (*Rosenthal et al., 2016*). Small insertion/deletions were interrogated for the presence of either short tandem repeat or microhomology at the breakpoints as described previously (*Nik-Zainal et al., 2016*). Complex indels were excluded from this analysis.

## Duplex mtDNA sequencing

Duplex Sequencing was performed as previously described (*Kennedy et al., 2014*) with the following modifications. Adapters were ligated to 100 ng of sheared DNA using the NEBNext Ultra II end-repair/dA-tailing and ligation kit (New England BioLabs) according to the manufacturer's instructions. The concentration of adapter ligated mtDNA was quantified by qPCR using the following primer sequences: 5'GTGACTGGAGTTCAGACGTGTGC and 5'-CCTCAACAGTTAAATCAACAA and a standard curve composed of ~400 bp of λ-phage DNA flanked with the same primer binding sites. After quantification, approximately $1 \times 10^6$ copies of fragmented adapter ligated mtDNA molecules werePCR amplified using KAPA HiFi DNA polymerase (Roche) as described in *Kennedy et al. (2014)*. The mtDNA was enriched using IDT xGen Lockdown probes (Integrated DNA Technologies) specific for human mtDNA and following the manufacturer's instructions. Each sample was sequenced with approximately $1 \times 10^7$ 150 bp paired-end reads on an Illumina NextSeq 500 platform.

The data were processed as previously described (*Kennedy et al., 2014*). To quantify the frequency of somatic mutations (*i.e.* not inherited), we used a clonality cutoff excluding any genomic positions with variant occurring at >1% or a depth of <100X and scoring each type of mutation only once at each genomic position. Additionally, all amino acid changes were also cataloged using software developed in-house and available upon request.

## Metabolomics

Metabolomic analysis of primary tumor and adjacent-normal tissue specimens in the RC13 dataset was conducted with Metabolon, Inc The protocol was identical to that described in *Hakimi et al. (2016)*, and is described in detail below.

### Sample acquisition

Tumors were obtained from patients after acquiring written informed consent and Memorial Sloan Kettering Cancer Center institutional review board approval. Partial or radical nephrectomies performed at Memorial Sloan Kettering Cancer Center (New York) were obtained by and stored at the MSK Translational Kidney Research Program (TKCRP). Samples were fresh frozen and stored at −80 degrees Celsius prior to metabolomic characterization. Clinical metadata including tumor pathologic and clinical stage, nuclear grade, metastatic status, and patient characteristics were recorded. All samples were reviewed by two expert genitourinary pathologists to confirm CCPAP or ccRCC histology.

### Sample preparation

The sample preparation process was carried out using the automated MicroLab STAR system from Hamilton Company. Sample preparation was conducted using a proprietary series of organic and

aqueous extractions to remove the protein fraction while allowing maximum recovery of small molecules. The resulting extract was divided into two fractions; one for analysis by LC and one for analysis by GC. Samples were placed briefly on a TurboVap (Zymark) to remove the organic solvent. Each sample was then frozen and dried under vacuum. Samples were then prepared for the appropriate instrument, either LC/MS or GC/MS.

## LC/MS

The LC/MS portion of the platform used Waters ACQUITY UPLC and a Thermo-Finnigan LTQ mass spectrometer, which consisted of an electrospray ionization (ESI) source and linear ion-trap (LIT) mass analyzer. The sample extract was split into two aliquots, dried, then reconstituted in acidic or basic LC-compatible solvents. Extracts reconstituted in acidic conditions were gradient eluted using water and methanol both containing 0.1% Formic acid, while the basic extracts, which also used water/methanol, contained 6.5 mM Ammonium Bicarbonate. One aliquot was analyzed using acidic positive ion optimized conditions and the other using basic negative ion optimized conditions in two independent injections using separate dedicated columns. The MS analysis alternated between MS and data-dependent $MS^2$ scans using dynamic exclusion.

## GC/MS

Samples analyzed with GC/MS were re-dried under vacuum desiccation for a minimum of 24 hr prior to being derivatized under dried nitrogen using bistrimethyl-silyl-triflouroacetamide (BSTFA). The GC column was 5% phenyl and the temperature ramp is from 40° to 300° C in a 16 min period. Samples were analyzed on a Thermo-Finnigan Trace DSQ fast-scanning single-quadrupole mass spectrometer using electron impact ionization. The instrument was tuned and calibrated for mass resolution and mass accuracy on a daily basis. The information output from the raw data files was automatically extracted as discussed below.

## Accurate mass determination and MS/MS fragmentation (LC/MS), (LC/MS/MS)

The LC/MS portion of the platform was based on a Waters ACQUITY UPLC and a Thermo-Finnigan LTQ-FT mass spectrometer, which had a linear ion-trap (LIT) front end and a Fourier transform ion cyclotron resonance (FT-ICR) mass spectrometer backend. For ions with counts greater than 2 million, an accurate mass measurement could be performed. Accurate mass measurements could be made on the parent ion as well as fragments. The typical mass error was less than five ppm. Ions with less than two million counts require a greater amount of effort to characterize. Fragmentation spectra (MS/MS) were typically generated in data dependent manner, but if necessary, targeted MS/MS could be employed, such as in the case of lower level signals.

## Data extraction and quality assurance

The data extraction of the raw mass spec data files yielded information that could loaded into a relational database and manipulated without resorting to BLOB manipulation. Once in the database the information was examined and appropriate QC limits were imposed. Peaks were identified using Metabolon's proprietary peak integration software, and component parts were stored in a separate and specifically designed complex data structure.

## Compound identification

Metabolites were definitively identified based on comparison to an in-house library of standards from Metabolon. Data on each of these standards included retention index, mass-to-charge ratio, and MS/MS spectra. Parameters for each of these three features for each compound in the metabolomic data were compared to analogous parameters in this library. As described in *Evans et al. (2009)*, compound identification was based on three criteria: retention index within 75 RI units of the proposed identification, mass to within 0.4 m/z, and MS/MS forward and reverse match scores.

## Data normalization

Each compound was corrected in run-day blocks by registering the medians to equal one (1.00) and normalizing each data point proportionately. The abundance of each compound was subsequently divided by the median abundance of that compound across all samples. As all statistical analyses of the data were non-parametric (see below), this median normalization was largely for the purposes of ease of analysis and data visualization.

## Data imputation

For instances when metabolite levels were below the level of detection, the level was imputed to be the lowest measured abundance of that compound across all samples.

## Determination of statistically significant changes in abundance

Non-parametric Mann-Whitney U tests were used for determination of differential abundance. All p-values were multiple-hypothesis corrected using the Benjamini-Hochberg procedure. The abundance reported for a given metabolite is the area for the peak that corresponds to the unadducted compound, as determined from the ion mass and comparison to the library entry. No adducts were merged into a single value.

## RC15 metabolomics

A Shimadzu Nexera X2 UHPLC combined with an AB Sciex TripleTOF 5600 of DuoSpray Ion Source was used in this study. The system was controlled by AB Sciex Analyst 1.7.1 instrument acquiring software. A Supelco Ascentis Express HILIC (7.5 cm ×3 mm, 2.7 µm) column was used with mobile phase (A) consisting of 5 mM NH4OAc and 0.1% formic acid; mobile phase (B) consisting of 98% acetonitrile, 2% 5 mM NH4OAc and 0.1% formic acid. Gradient program: mobile phase (B) was held at 90% for 0.5 min and then increased to 50% in 3 min; then to 99% in 4.1 min and held for 1.4 min before returning initial condition. The column was held at 40°C and 8 µl of sample was injected into the LC–MS/MS with a flow rate of 0.4 ml/min. Calibrations of TOFMS were achieved through reference APCI source with average mass accuracy of less than five ppm. Key MS parameters: Mass range 50–800 Da, declustering potential (DP) −80V, collision energy (CE) −30 eV with 15 eV spread, ion Release Delay (iRD) 30 V, ion Release Width (iRW) 15V: source parameters: curtain gas 30 psi, GS1 50 psi, GS2 50 psi, source temperature 500 C, ISVF 4000 volts for negative acquisition. TOF scan followed by 49 high sensitivity TOF MS/MS product ion scans were set on the each cycle of MS method. Data Processing Software included Sciex PeakView 2.2, MasterView 1.1, LibraryView (64 bit) and MultiQuant 3.0.2.

## t-SNE Analaysis

Log-2 transformed metabolomics data was analyzed using the rtsne R package with perplexity set to 5. Error were observed to stabilize after approximately n = 1000 iterations.

## Transcriptomic analysis

Gene expression profiles were downloaded from firebrowse.org for the relevant TCGA studies. Differential expression was completed using the limma voom package, and gsea was completed using gseapy (*Kuleshov et al., 2016*).

microRNA analysis was done using batch-corrected miRNA profiles from *Chen et al. (2016)*. miRNAs with expression greater than 20 were evaluated. Differential expression was determined with t-test using log2-transformed expression values. False Discovery Rates estimated using method of *Storey and Tibshirani (2003)*.

## Methylation analysis

DNA Methylation for TCGA samples was downloaded from the UCSC Xena Browser (http://xena.ucsc.edu/). We used level three data that had been acquired using the Illumina HumanMethylation450K platform and had been pre-processed by TCGA using their standard protocols. Data was available for 219 solid tumors and 199 adjacent non-cancer samples. We discarded all the probes that interrogated locations in chromosomes X and Y, as well as all probes that were masked as NA ('Not Available') for more than 90% of the samples. For principal components analysis, we used the

beta values for the set of 59,577 probes that interrogated locations within CpG islands (excluding shores and shelves), as defined in the Illumina documentation for the array. These data were standardized and subsequently principal components analysis was applied.

## Immunohistochemistry

Immunohistochemical stains using the antibodies TOM20 (rabbit polyclonal, Santa Cruz Biotech., Dallas, TX; dilution 1:100), MTCO1 (mouse monoclonal, Abcam, Cambridge, MA; dilution 1:2000), and Anti-8 Hydroxyguanosine (mouse monoclonal, Abcam, Cambridge, MA; dilution 1:400) were performed on formalin-fixed, paraffin-embedded tissues using the manufactures' protocols.

The stained sections were scanned using the Aperio ScanScope XT systems (Aperio Technologies, Vista, CA) at 20X magnification. Immunohistochemical expression was quantified in all digital slides with either nuclear (Anti-8 Hydroxyguanosine) or cytoplasmic (TOM20 and MTCO1) algorithms using ImageScope analysis software (version 12; Aperio Technologies, Inc) and an H–score was calculated (*Luna, 1968*).

## Histochemistry

PAS/PAS diastase (PASD) and Oil Red O staining was performed using the standard methods (*Luna, 1968*; *Masson, 1929*). While frozen tissue samples were used for Oil Red O, fixed paraffin-embedded tissue was utilized for PAS and PASD.

## Additional information

### Competing interests
Steve Stirdivant: This author is a former employee of Metabolon, Inc. The other authors declare that no competing interests exist.

### Funding

| Funder | Grant reference number | Author |
| --- | --- | --- |
| Damon Runyon Cancer Research Foundation | Dale F. Frey Award for Breakthrough Scientists, DFS-09-14 | Costas A Lyssiotis |
| V Foundation for Cancer Research | Junior Scholar Award, V2016-009 | Costas A Lyssiotis |
| Sidney Kimmel Foundation for Cancer Research | Kimmel Scholar Award, SKF-16-005 | Costas A Lyssiotis |
| National Institutes of Health | DK097153 | Costas A Lyssiotis |
| Charles Woodson Research Fund | | Costas A Lyssiotis |
| The UM Pediatric Brain Tumor Initiative | | Costas A Lyssiotis |
| University of Michigan | Program in Chemical Biology - Graduate Assistance in Areas of National Need (GAANN) award | Daniel Kremer |

| National Cancer Institute | P30 CA008748 | Jianing Xu<br>Ed Reznik<br>Gunes Gundem<br>Philip Jonsson<br>Judy Sarungbam<br>Anna Bialik<br>Francisco Sanchez-Vega<br>Jozefina Casuscelli<br>Nikolaus Schultz<br>Yiyu Dong<br>Paul Russo<br>Jonathan A Coleman<br>Elli Papaemmanuil<br>Ying-Bei Chen<br>Victor E Reuter<br>Chris Sander<br>Satish K Tickoo<br>A Ari Hakimi |
| --- | --- | --- |
| National Cancer Institute | P30 CA046592 | Costas A Lyssiotis |
| Sidney Kimmel Foundation for Cancer Research | | A Ari Hakimi |
| American Urological Association | Research Scholar Award | A Ari Hakimi |

The funders had no role in study design, data collection and interpretation, or the decision to submit the work for publication.

## Author contributions

Jianing Xu, Conceptualization, Data curation, Formal analysis, Validation, Investigation, Visualization, Writing—original draft, Writing—review and editing; Ed Reznik, Conceptualization, Data curation, Software, Formal analysis, Validation, Investigation, Writing—original draft, Writing—review and editing; Ho-Joon Lee, Formal analysis, Software, Data curation, and Investigation, Provided critical feedback during drafting of the manuscript and approved its final version; Gunes Gundem, Philip Jonsson, Formal analysis, Provided critical feedback during drafting of the manuscript and approved its final version; Judy Sarungbam, Anna Bialik, Francisco Sanchez-Vega, Jake Hoekstra, Li Zhang, Peter Sajjakulnukit, Daniel Kremer, Zachary Tolstyka, Wenjing Su, Investigation, Provided critical feedback during drafting of the manuscript and approved its final version; Chad J Creighton, Software, Formal analysis, Investigation; Jozefina Casuscelli, Resources, Data curation; Steve Stirdivant, Scott R Kennedy, Formal analysis, Investigation; Jie Tang, Data curation, Formal analysis, Investigation; Nikolaus Schultz, Yiyu Dong, Paul Russo, Jonathan A Coleman, Elli Papaemmanuil, Victor E Reuter, Chris Sander, Resources, Provided critical feedback during drafting of the manuscript and approved its final version; Paul Jeng, Visualization, Provided critical feedback during drafting of the manuscript and approved its final version; Emily H Cheng, Resources, Funding acquisition; Ying-Bei Chen, Resources, Investigation; James J Hsieh, Resources, Formal analysis, Supervision, Funding acquisition; Costas A Lyssiotis, Conceptualization, Data curation, Formal analysis, Investigation, Writing—review and editing; Satish K Tickoo, Conceptualization, Resources, Data curation, Formal analysis, Funding acquisition, Investigation; A Ari Hakimi, Conceptualization, Resources, Data curation, Formal analysis, Supervision, Funding acquisition, Investigation, Writing—original draft, Project administration, Writing—review and editing

## Author ORCIDs

Ed Reznik https://orcid.org/0000-0002-6511-5947
Ying-Bei Chen http://orcid.org/0000-0001-5207-3648
A Ari Hakimi http://orcid.org/0000-0002-0930-8824

## Ethics

Human subjects: Frozen samples and genomic data were acquired through MSKCC IRB approved tissue protocol 06-107.

Decision letter and Author response
Decision letter https://doi.org/10.7554/eLife.38986.024
Author response https://doi.org/10.7554/eLife.38986.025

# Additional files

**Supplementary files**
• Transparent reporting form
DOI: https://doi.org/10.7554/eLife.38986.022

**Data availability**

The new data generated in this study is primarily tumor/germline sequencing of primary human tumor specimens, and constitutes human subject data. To protect the privacy of the human subjects, we have included somatic mutation calls in Figure 4 Source Data 1, but have withheld germline information. Source data have been provided for Figures 1-5. Controlled access for TCGA sequencing data (RNA-sequencing and whole exome sequencing of CCPAP tumors) are available via GDC commons data portal (https://gdc.cancer.gov/) by querying the 5 CCPAP sample IDs (BP-4760, BP-4784, BP-4795, DV-5567, BP-4177). Data from the The Cancer Genome Atlas Pan-Cancer Analysis Project related to this studied can be downloaded directly from firebrowse.org at the url http://gdac.broad-institute.org/runs/stddata__2016_01_28/data/KIPAN/20160128/.

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
