## [Decision Letter]

Thank you for submitting your article "Abnormal oxidative metabolism in a quiet genomic background underlies clear cell papillary renal cell carcinoma" for consideration by *eLife*. Your article has been reviewed by three peer reviewers, including Ralph DeBerardinis as the Reviewing Editor and Reviewer #1, and the evaluation has been overseen by Sean Morrison as the Senior Editor. The following individual involved in review of your submission has agreed to reveal their identity: Christian Frezza (Reviewer #2).

The reviewers have discussed the reviews with one another and the Reviewing Editor has drafted this decision to help you prepare a revised submission.

Summary:

Kidney tumors of multiple histological classes have evidence of perturbed mitochondrial metabolism. In some forms of kidney cancer, these alterations result from recurrent driver mutations such as the loss of VHL, fumarate hydratase, or components of the succinate dehydrogenase complex. Here the authors use metabolic and molecular analyses to characterize a distinct type of renal cancer, clear cell papillary renal cell carcinoma (CCPAP), which shares some histological characteristics with the much more common clear cell renal cell carcinoma (ccRCC), but lacks the usual mutations observed in those tumors. A metabolomic analysis revealed numerous changes that distinguish CCPAP from either adjacent kidney tissue or ccRCC tumors; these included enhanced markers of oxidative stress and a large increase in sorbitol and other sugar alcohols. Other molecular analyses revealed marked suppression of transcripts related to mitochondrial metabolism, including transcripts from the nuclear genome and the mitochondrial DNA (mtDNA). The mtDNA copy number was also suppressed relative to kidney and ccRCC. Surprisingly, despite these consistent molecular and metabolic features, exome and genome sequencing failed to identify recurrent mutations in a modestly-sized cohort of CCPAP tumors. The authors conclude by arguing for non-genomic profiling including metabolomics as a means to classify tumors. Overall, the paper provides a thorough analysis of a rare but interesting subclass of tumors that may ultimately inform us about the role of metabolic alterations in tumor initiation or progression.

Essential revisions:

1) The metabolomic analysis is thorough, but based entirely on semi-quantitative measurements. It would improve the analysis if the authors could definitively quantify some of the more important metabolites, particularly sorbitol, which the authors define as a key metabolite in these tumors. The authors should also describe how they differentiated sorbitol from other sugars with the same mass.

2) The authors use the metabolomics data to infer a shift away from oxidative metabolism, but classical markers of impaired respiration (e.g. elevated lactate) are not observed in the heat map. This warrants discussion.

3) It is unclear whether the data really demonstrate a change in the NAD+/NADH ratio in CCPAP. This ratio should impact many metabolic reactions, so if possible the authors should highlight other metabolites from their data set in addition to sorbitol consistent with an impaired ratio. Using metabolomics to calculate the NAD+/NADH ratio is potentially fraught with artifacts, so an orthogonal method should be used to validate the ratio. Also, it is curious that the ratio is similar between ccRCCs and CCPAP, but CCPAPs accumulate much more sorbitol. Is sorbitol elevation a general consequence of mitochondrial dysfunction, or are there additional factors specific to CCPAP that favor soribitol accumulation?

4) The reviewers raised several questions about the sorbitol labeling experiment in Figure 6. First, the overall abundance of sorbitol in all these conditions is low; in the mock condition, the level is low enough that it is difficult to rule out modest labeling of the pool. Conditions should be altered (e.g. increasing the cell number) to yield a larger sorbitol pool for analysis. Second, this experiment should be extended beyond a single 16 hour time point. Analyzing sorbitol enrichment at several time points to thoroughly assess steady-state labeling, preferably together with quantitation of labeled sorbitol abundance, would better make the case for increased flux through this pathway in the context of electron transport chain inhibition. Third, interpreting the labeling data requires some knowledge of the rates of fructose uptake and the fractional labeling of fructose in the cells. Fourth, the relatively modest labeling of sorbitol argues that other carbon sources (e.g. glucose) contribute to the pool. This should be evaluated by performing similar experiments with labeled glucose.

5) The overabundance of sorbitol in vivo is interpreted as excess synthetic flux, but it is also possible that CCPAP tumors reduce utilization of polyol products. Any data to differentiate these possibilities should be included in the paper, or at least the authors should acknowledge this possibility in the Discussion.

6) Does depletion of mtDNA in CCPAP accompany reduced mitochondrial mass? If tissue is available, electron microscopy to estimate the mitochondrial mass content would help address this question.

7) There were concerns with the claim that CCPAP tumors experience oxidative stress. As with the NAD+/NADH ratio, it is difficult to obtain reliable measurements of GSH and GSSH using metabolomics. 8-oxo-dG may report oxidative stress, but it also indicates DNA damage. The authors should better assess oxidative stress in these tumors, or soften the argument that CCPAP and ccRCC tumors differ in oxidative stress.

---

## [Author Response]

Essential revisions:1) The metabolomic analysis is thorough, but based entirely on semi-quantitative measurements. It would improve the analysis if the authors could definitively quantify some of the more important metabolites, particularly sorbitol, which the authors define as a key metabolite in these tumors. The authors should also describe how they differentiated sorbitol from other sugars with the same mass.

We agree with the feedback from the reviewers. We have taken several different approaches to confirm the validity of our metabolite measurements, see below.

a) Using a distinct metabolomic platform (RC15, see Materials and methods) we ran a standard curve with sorbitol to determine the absolute concentration of sorbitol in several CCPAP tumor samples, including both high- and low sorbitol CCPAP tumors (Figure 2—figure supplement 1B). The levels of sorbitol in these tumors accumulate to on the order of 100 ug/mg of tissue, with significant differences between both high-sorbitol/low-sorbitol and tumor/normal samples. This data was in good agreement with the Metabolon measurements of sorbitol(Figure 2—figure supplement 1C). As part of this analysis, we also analyzed 3 additional CCPAP tumors in RC15 (samples 225, 350, and 396), and observed similarly high levels of sorbitol as of the samples profiled in RC13.

b) With regard to the question of disambiguating sorbitol from other sugars with the same mass, we returned to the original Metabolon data. As shown in a newly-included Figure 2—source data 2, sorbitol had a distinct retention index from other sugar alcohols with the same mass (e.g. 1843 for sorbitol vs. 1839 for mannitol). This is indicated in the first paragraph of the subsection CCPAP is metabolically characterized by alterations to sorbitol metabolism and oxidative stress pathways”.

c) We further evaluated whether data from the 6 samples profiled both by Metabolon (RC13) and the Michigan group (RC15) exhibited reproducible patterns. As shown in Figure 2—figure supplement 1C, for the majority of metabolites profiled by both of the platforms, we observed similar patterns of abundance. However, some metabolites did show differences in abundance, which may arise from both biological variation (physically distinct areas of the tumor were profiled) as well as technical noise/error. This result is described in the aforementioned paragraph. As noted in point a above, sorbitol abundance was highly similar between the Michigan and Metabolon data.

2) The authors use the metabolomics data to infer a shift away from oxidative metabolism, but classical markers of impaired respiration (e.g. elevated lactate) are not observed in the heat map. This warrants discussion.

We agree with the reviewers and have now discussed this in the fourth paragraph of the Discussion. Briefly, we believe that, because these tumors are slow growing and largely indolent, they may not up-regulate glycolytic flux to the same extent as other, more malignant cancers.

3) It is unclear whether the data really demonstrate a change in the NAD+/NADH ratio in CCPAP. This ratio should impact many metabolic reactions, so if possible the authors should highlight other metabolites from their data set in addition to sorbitol consistent with an impaired ratio. Using metabolomics to calculate the NAD+/NADH ratio is potentially fraught with artifacts, so an orthogonal method should be used to validate the ratio. Also, it is curious that the ratio is similar between ccRCCs and CCPAP, but CCPAPs accumulate much more sorbitol. Is sorbitol elevation a general consequence of mitochondrial dysfunction, or are there additional factors specific to CCPAP that favor soribitol accumulation?

The reviewers are correct. To clarify whether the NADH/NAD ratio was substantially changed in CCPAP relative to normal tissue, we used the Recon2 human metabolic reconstruction to identify all metabolic reactions in the cell which are dependent on oxidation of NADH/reduction of NAD. We obtained a list of all participating metabolites in these reactions, and then intersected that list with the metabolites measured in our study. We retained all reactions for which we measured all substrates and products. Because several of the central metabolites participating in reactions using NADH/NADH were not quantified in our dataset (e.g. pyruvate, alpha-ketoglutarate, oxaloacetate, dihydroxyacetone phosphate), a number of canonical NAD-dependent reactions were not retained. However, we were left with a list of 5 reactions (related to xanthine metabolism, sorbitol metabolism, ascorbate metabolism, and uracil metabolism). For each of these reactions, we calculated the ratio of the abundances of the more oxidized metabolite to the abundance of the more reduced metabolite (specifically, fructose/sorbitol, uracil/5,6 dihydrouracil, xanthine/hypoxanthine, urate/xanthine). In conditions of elevated NADH/NAD, we would expect the chemical equilibrium to shift these ratios lower. Consistent with this, of these 5 ratios, 4 exhibit a shift in metabolite levels consistent with an elevated NADH/NAD ratio (i.e. a depletion in the ratio). This is shown in Figure 2—figure supplement 2 and mentioned in the last paragraph of the subsection “CCPAP is metabolically characterized by alterations to sorbitol metabolism and oxidative stress pathways”.

To determine if sorbitol accumulation has been observed in other contexts associated with mitochondrial dysfunction, we reviewed the literature. A review of metabolomic changes associated with (germline) mitochondrial disease discussed the pattern of metabolic changes associated with various mitochondrial diseases. Interestingly, only one report described an accumulation of sorbitol in patients with PEO, and even in this case, it was not discussed in the main text. Based on this, we speculate that sorbitol accumulation in CCPAP is the outcome of somatic genetic changes specific to the tumor and the cellular lineage. We describe this in the third paragraph of the Discussion.

4) The reviewers raised several questions about the sorbitol labeling experiment in Figure 6. First, the overall abundance of sorbitol in all these conditions is low; in the mock condition, the level is low enough that it is difficult to rule out modest labeling of the pool. Conditions should be altered (e.g. increasing the cell number) to yield a larger sorbitol pool for analysis. Second, this experiment should be extended beyond a single 16 hour time point. Analyzing sorbitol enrichment at several time points to thoroughly assess steady-state labeling, preferably together with quantitation of labeled sorbitol abundance, would better make the case for increased flux through this pathway in the context of electron transport chain inhibition. Third, interpreting the labeling data requires some knowledge of the rates of fructose uptake and the fractional labeling of fructose in the cells. Fourth, the relatively modest labeling of sorbitol argues that other carbon sources (e.g. glucose) contribute to the pool. This should be evaluated by performing similar experiments with labeled glucose.

We agree with the reviewers, and revised the design of our labeling experiments. We performed experiments by using more cells (from 10 cm dishes instead of 6 well dish), labeling with either U13C-fructose or U13C-glucose, and harvesting at 5 additional time points (0h, 2h, 4h, 8h, 24h). We have included several of the key data points in Author response images 1-3, from which we have concluded that (1) both glucose and fructose contributed modestly to flux into sorbitol, and (2) that glucose/fructose-derived flux into sorbitol is not dependent on inhibition of electron transport chain activity (at least, in our model system). These data are shown in Author response image 1.

**Author response image 1. respfig1:** Sorbitol Labeling from Glucose or Fructose in Mock Condition (**A**), Antimycin A Condition (**B**), and Rho-Zero Condition (**C**). Left panel indicates total ion count. Right panel indicates labeling proportions, corrected for abundance of natural isotopes. Leftmost data of each panel is glucose labeling, rightmost data of each panel is fructose labeling.

We have attempted to determine the cause of the discrepancy between these findings and our initial findings. It is possible that the extremely small pool of analyzed sorbitol led to an error in our labeling quantification. As the reviewers correctly suggested, increasing cell number and using several time points ameliorated this issue. As a result of these experiments, we have removed the data associated with the sorbitol labeling experiment from the manuscript.

5) The overabundance of sorbitol in vivo is interpreted as excess synthetic flux, but it is also possible that CCPAP tumors reduce utilization of polyol products. Any data to differentiate these possibilities should be included in the paper, or at least the authors should acknowledge this possibility in the Discussion.

We acknowledge that it is also possible the reduced utilization of sorbitol products can lead to the overabundance of sorbitol in vivo. We added a discussion of this in the main text:

“While CCPAP tumors exhibit a uniquely high abundance of sorbitol, this should not be misconstrued as an indicator that flux through the polyol pathway has increased. […] This suggests that, if the accumulation of sorbitol is due to decreased utilization of metabolites downstream of sorbitol, the “choke-point” most likely occurs after fructose and/or sorbitol-6-phosphate.”

6) Does depletion of mtDNA in CCPAP accompany reduced mitochondrial mass? If tissue is available, electron microscopy to estimate the mitochondrial mass content would help address this question.

Thank you for asking us to clarify. The clinical management of RCC (and the rarity of CCPAP tumors) makes it difficult to obtain CCPAP tissue for electron microscopy; preoperative biopsies are generally not done in the clinical care of RCC patients, meaning that the histology is only identified after the tumor has been removed and not amenable to electron microscopy. However, we have several additional pieces of data that we use to infer that mitochondrial mass is indeed reduced in CCPAP tumors relative to normal tissue:

a) We immunohistochemically assess the levels of TOM20, an outer mitochondrial membrane protein, in CCPAP and adjacent normal tissue. We consistently find that TOM20 is reduced in CCPAP relative to normal tissue. However, the degree of depletion was smaller than when using MT-CO1 as the immunohistochemical marker (Figure 5).

b) We examined the expression levels of nuclear-DNA-encoded genes localizing to mitochondria. These genes are broadly down-regulated in CCPAP relative to normal kidney tissue in the TCGA (subsection “CCPAP exhibits dysfunctional mitochondrial respiration associated with depletion of 219 mitochondrial DNA and RNA”, last paragraph, Figure 4—figure supplement 1A). Specifically, of the 852 genes in the Mitochondrion GO geneset and expressed at a minimal level in CCPAP tumors, 411 were downregulated and 99 were upregulated in CCPAP tumors compared to normal tissue.

Together, these two pieces of data suggest that mitochondrial mass is indeed lower in CCPAP tumors, and (secondarily) that one of the most significant contributors to this reduction in mass is depletion of mtDNA itself.

7) There were concerns with the claim that CCPAP tumors experience oxidative stress. As with the NAD+/NADH ratio, it is difficult to obtain reliable measurements of GSH and GSSH using metabolomics. 8-oxo-dG may report oxidative stress, but it also indicates DNA damage. The authors should better assess oxidative stress in these tumors, or soften the argument that CCPAP and ccRCC tumors differ in oxidative stress.

This is a good point. We have made several revisions to the manuscript in this regard:

a) Abundant data from the last 5 years have demonstrated that different mutation-generated biological processes leave distinct signatures of DNA damage, e.g. the preference for C>A mutations at TpCpT trinucleotides in POLE-mutated tumors. Oxidative stress similarly induces preferential C>A mutations. Using our whole genome and whole exome sequencing data, we analyzed mutation signatures from each our genomically profiled tumors. We observed that the majority of mutations were C>T, which is not consistent with the expected bias towards C>A mutations in conditions of high oxidative stress (subsection “Immunohistochemical Validation of the Molecular Phenotype of CCPAP”, third paragraph, Figure 5—figure supplement 2). We interpret this to mean that 8-oxo-dG is reporting high levels of oxidative stress which are being sufficiently handled by the cell’s stress response (e.g.elevated GSH levels) so as to not generate excess oxidative stress-associated somatic mutations. We elaborate on this in the aforementioned paragraph.

b) Data from our lab and many others have repeatedly indicated that apparently all histologies of RCC (clear cell, chromophobe, oncocytoma, CCPAP; perhaps not papillary RCC, there is only sparse data) have elevated GSH and GSSG levels. Therefore, it is unlikely that this is a distinguishing molecular feature of CCPAP, and we agree that our conclusions should be softened in the manuscript. We have revised the manuscript to indicate this in the aforementioned paragraph.